# Thermodynamic Conditions during August 2022 in Catalonia: The Monthly Record of Hail Days, Hail Size and the Differences in the Climatic Values

**Tomeu Rigo**

Servei Meteorològic de Catalunya, C/ Dr. Roux, 80, 08017 Barcelona, Spain; tomeu.rigo@gencat.cat

**Abstract:** The hailstorm of 30 August 2022 in the NE of Catalonia (NE of the Iberian Peninsula) produced the largest hail size, with diameters exceeding 10 cm. Furthermore, hail occurrence exceeded 2 cm in fourteen days and 4 cm in seven days during August 2022. The size and the days number constituted new records in Catalonia for at least the last 30 years. The analysis has compared the thermodynamic values derived from the sounding of Barcelona with the climatic data for 1998–2022 (25 years of data). This fact has allowed the selection and evaluation of different thermodynamic parameters' behaviour during hail days in Catalonia. In this research, the precipitable water mass provided the best results as a hail forecaster. Second, the study has evaluated if the different parameters have a significant trend during the study period. The answer is yes in all cases, but some parameters presented a stepped rise while others increased smoothly. Finally, the research has analysed if the parameter values during August 2022 were extraordinary compared with the rest of the period. In this case, the thermodynamic parameters nature was well correlated with the hail size and occurrence maximums of August 2022.

**Keywords:** giant hail; radiosonde; convection; thermodynamic indexes; Catalonia

## 1. Introduction

The event of 30 August 2022 was the first giant hail case in Catalonia recorded in the last thirty years in the region, according to the Severe Weather Data Base of the Servei Meteorològic de Catalunya [1,2]). The reports collected in this database correspond to different sources (official spotters, automatic weather stations, hail-managing companies, and social networks). Each element has been manually validated considering remote sensing data (radar and lightning) and includes the location, the date and time, the magnitude of the event, and the source. Rigo and Farnell [3] showed that for the 2013–2021 period, some averaged values were 90 hail observations per year, 4 cm of maximum yearly hail size, and 18 hail days per year.

In recent years, other regions of the Mediterranean (such as Greece [4] or Italy [5]) have registered similar events, with hail stones exceeding 10 cm. These registers are extraordinary in these regions, based on the research of Púčik et al. [6]. Some authors have studied the future trends of large and very large hail around the world, considering database records or the predicted scenarios, including global warming. These studies [7–10] have concluded that trends are very different depending on the region, but most of them coincide in a future increase in the severe hailstorms.

The general environments favourable to large hail development are well-known for some decades [7]. Hailstorms develop in environmental conditions that include suportive updraft strength (large values of CAPE—Convective Available Potential Energy—or mid-level temperature lapse rate [11,12], vertical wind shear between surface and low or mid-levels [13,14], the freezing level height [15], or the amount of humidity available [16]. However, Allen et al. [7] stated that all those ingredients depend on the region and the

synoptic conditions, agreeing with previous research performed in the NE of the Iberian Peninsula or the Southern France [8,17,18].

Johnson and Sundgen (2014) [16] divided the sounding-derived parameters into two types: wind-related and thermodynamic. Knight and Knight (2001) [19] showed an inverse relation between maximum hail size and storm depth shear. [16] confirmed this assumption through large-hail database. However, wind-related parameters did not provide good results in categorizing hail size in thunderstorm environments. Moving to the thermodynamic parameters, Groenemeijer and Van Delden (2007) [20] found a good correlation between hail size and large CAPE values. However, other parameters (such as shear) should be favourable to favour hail occurrence. Edwards and Thompson (1998) [21] indicated a series of assumptions related to hail occurrence in thunderstorms: (a) the higher the Equilibrium Level (EL), the larger the hail size because the embryos have reached higher altitudes; (b) the lower is the Freezing Level (FRZL), the larger the hail size because of the lower exposition of hailstones to melting; and (c) the higher the CAPE, the larger the hail size because the updraft strength increases. However, they could not confirm any of the previous hypotheses. Finally, Johnson and Sundgen (2014) [16] found similar results, especially for discriminating between hail size categories. In this way, they suggested integrating thermodynamic and wind-derived parameters with micro-physics in the hail-storm initiation, considering the difficulties of applying them in operational tasks.

The use of radio-sounding data for analysing the thermodynamic conditions of the atmosphere has a long history in Catalonia. Gibergans [22] used the data from different stations for analysing heavy rainfall events occurring during Autumn. Moving to severe weather, Rodríguez [23] studied the favourable conditions for tornadoes and waterspouts development in Catalonia. Finally, Farnell [2,24] established the basis of the necessary ingredients for the occurrence of hail and large hail in the region. Furthermore, the authors proposed some thresholds for the different parameters: maximum vertical updraft of 50 m s$^{-1}$, freezing level around 3000 m ASL, Lifted Index (LI) of $-5$ °C, vertical shear in the 0–6 km level of 15 m s$^{-1}$, and Precipitable Water Content (PWC) of 35 mm. Other complementary values are CAPE of 2000 J kg$^{-1}$ or humidity between surface and 850 hPa below 70%. These indices and thresholds have been the starting point of the current analysis.

The summer of 2022 was historical in Catalonia for two reasons: the number of severe hail events (diameter $\geq$ 2 cm) [3] and the largest hail stone event that has occurred in the region in at least the last 30 years [25]. The main goal of this analysis is to validate the previously presented parameters, which can improve the forecasting of that kind of event. Furthermore, two secondary objectives have been the evaluation of these parameters for the specific case of August 2022 (the month of the hail record) and, finally, the parameters trend determination along the period of study (1998–2022).

## 2. Materials and Methods

The research has consisted of some parameters analysis derived from the radio sounding of Barcelona (World Meteorology Organization code 08190, marked with a red triangle in Figure 1). The sounding launching started in the Autumn of 1997 to cover the Northeastern Iberian Peninsula. Sounding data used are from twice a day, for 00.00 and 12.00 UTC. This region is prone to be affected by different types of adverse weather (hail [2], heavy rain and floods [26], snow at low altitude [27], tornadoes [28], etc.). Before, the understanding capacity of the thermodynamic conditions over Catalonia was poor (Gibergans-Báguena and Llasat [22]).

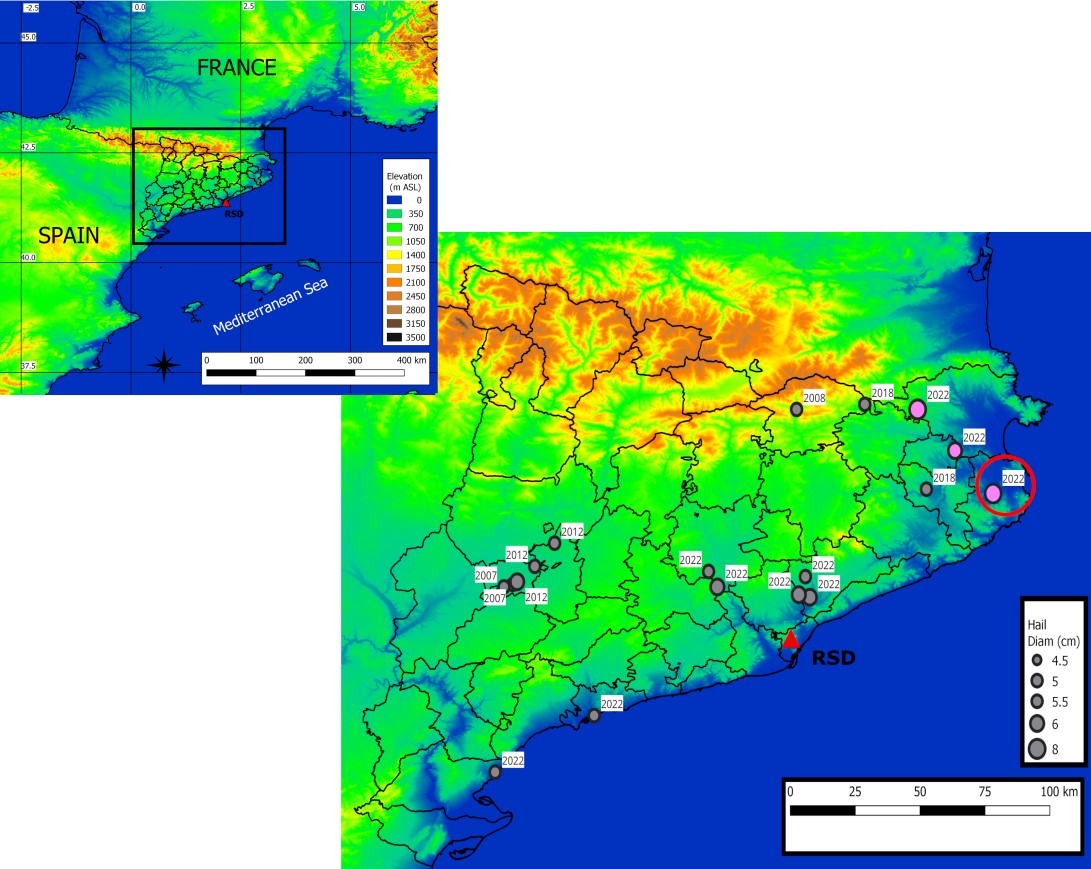

**Figure 1.** Physical map of the study area: general view of Catalonia (NE of the Iberian Peninsula) and surroundings (**top-left**), and zoomed view (included in the black rectangle) focused on the studied region (**bottom-right**). The red triangle indicates the location of the sounding launching base (in Barcelona, labeled with RSD). Grey dots in the bottom-right panel mark the largest hail that occurred in Catalonia since 2004 (labels correspond to the occurrence year, except the pink ones that make reference to the 30 August 2020 event). The red circle marks the hail record in Catalonia.

The data have been analysed through the R package thundeR [29,30], considering some steps of the quality control developed by Abellan et al. [31]. This technique consisted mainly in detecting anomalous values (exceeding the top and bottom outliers for the different magnitudes or incoherent vertical evolution of the parameters). However, first, it was necessary to eliminate or repair some sounding files with anomalous characteristics (corrupted data or incomplete values). The values' reparation consists of two steps. First, it has applied an automatic algorithm that interpolates the surrounding values. The first possibility is that the interpolated value is similar to the erroneous one but coherent with the neighbour data (for instance, a negative value between two positive data or the lack of a digit—e.g., "6.7" instead of "60.7"). In this case, the technique directly replaces the value. On the contrary, if the algorithm does not recognize the interpolated value as a good chance for substituting the anomalous data, then it labels the register for applying a manual replacement. This second step implies a human recognition of the data and changing the value directly. Figure 2 shows the quality control process steps.



QC0: detection of soundings with scarce observations

QC1: identification of incomplete files

QC2: labelling of files with anomalous values (T, Td, Wd, Ws)

**Figure 2.** Quality control (QC) process for all the sounding files analysed during the research. The step QC2 is based on [31]. T means temperature, Td is dew point temperature, Wd is wind direction, and Ws is wind speed.

The radiosondes have been analysed through the thundeR package, allowing us to obtain representative plots of the atmospheric state (see the examples for the 00 and 12 UTC of 30 August 2022 in Figure 3). Furthermore, the same package provides a series of variables that give quantitative and qualitative information about atmospheric thermodynamics. The variables itemized in Table 1 have been used in this research. The table contains the description and the helpful (forecasting purpose) of each variable. Additionally, it has estimated EL and WMAX (maximum updraft speed) in two ways, considering the classic way (surface-based, SB) and through the virtual temperature (most unstable, MU). Different authors [32,33] have shown the advantages of the second option, the first from the theoretical point of view and the second from an operational perspective. However, the second reference concluded that combining both calculating ways is optimal.

**Table 1.** Different thermodynamic variables derived from the sounding and considered in this research.

| Variable | Description/Purpose |
|---|---|
| MUEL (m AGL) | Height of the equilibrium level, derived from the most-unstable parcel (highest theta-e between surface and 3 km AGL). Determine the vertical thunderstorm MU development. |
| MUWMAX (m s$^{-1}$) | the maximum updraft speed in a thunderstorm (a square root of two times CAPE), derived from the most-unstable parcel (highest theta-e between surface and 3 km AGL). Calculate the strength of the updraft in the most-unstable case. |
| SBEL (m AGL) | Height of the equilibrium level, derived from the surface-based parcel. Determine the vertical thunderstorm SB development. |
| SBWMAX (m s$^{-1}$) | The maximum updraft speed in a thunderstorm (a square root of two times CAPE), derived from the surface-based parcel. Estimate the strength of the updraft in the most-unstable case. |
| PWM (mm) | Precipitable water mass (entire column). Equivalent to Precipitable Water Content. Measure the water quantity available. for freezing. |
| FRZL (m AGL) | Height of freezing level (0 °C) as a first available level counting from the surface. Determine the melting path length of hailstones. |

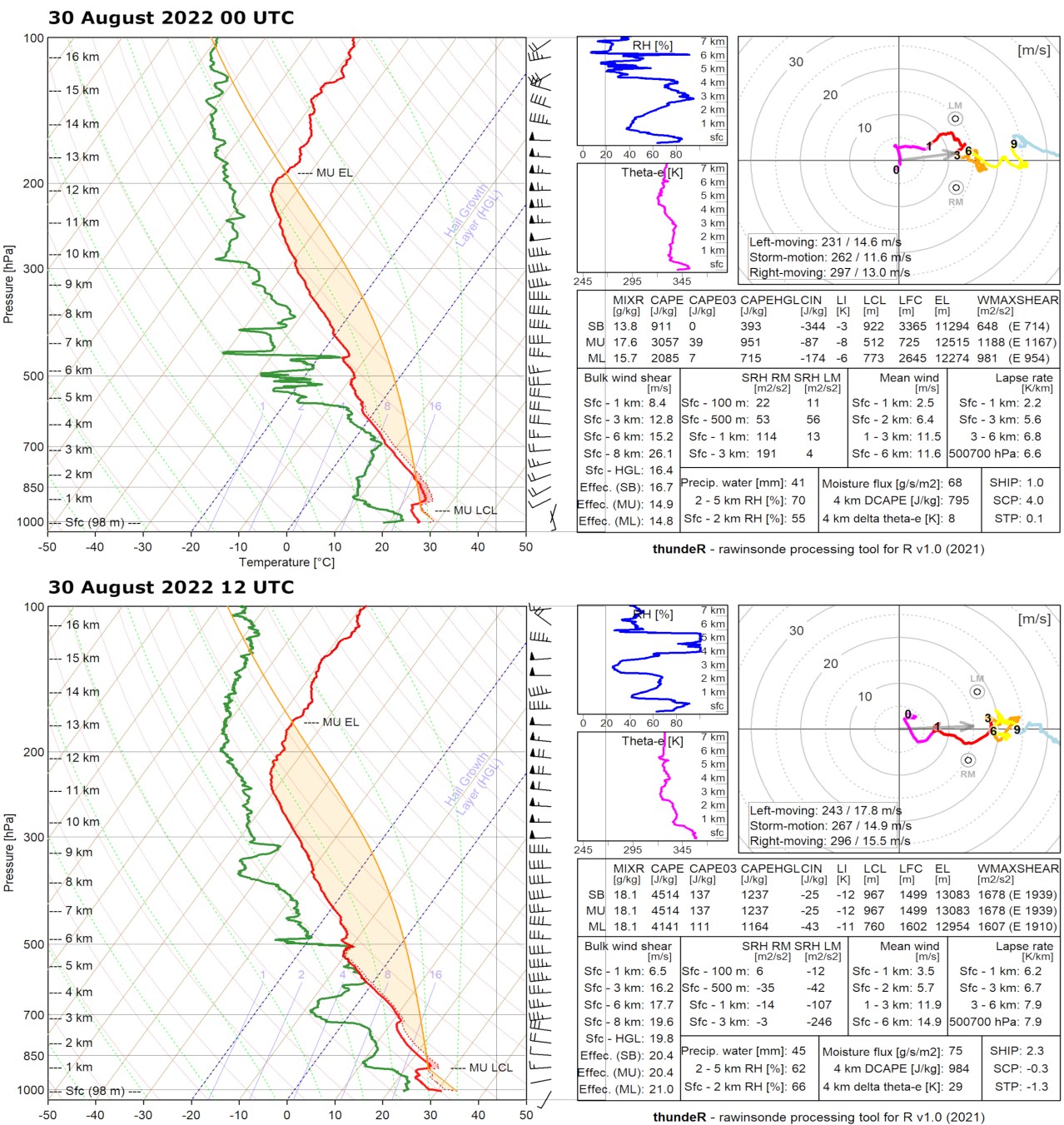

**Figure 3.** Sounding profiles and other thermodynamic features obtained from the thunder software [30] for the episode of the 30 August 2022 (top: at 00 UTC, bottom: at 12 UTC). For more information about the composite plot, see https://bczernecki.github.io/thundeR/articles/getstarted.html (accessed on 15 July 2023).

The hail days detection was made using the Severe Weather Database registers (hereafter SWDB) of the Meteorological Service of Catalonia (hereafter SMC). The database includes hail events, wind gusts associated with thunderstorms, and tornadoes in Catalonia from 2004 [34]. The maximum hail sizes derived from the SWDB included episodes in 2007 (5 cm), in 2012 and 2014 (7 cm), and at the end of August 2022 (10 cm, the absolute record in the region, see Figure 4). In all cases, the time of the hail event was between 15 and 18 UTC, the most usual in the region of study according to Farnell and Rigo [35].

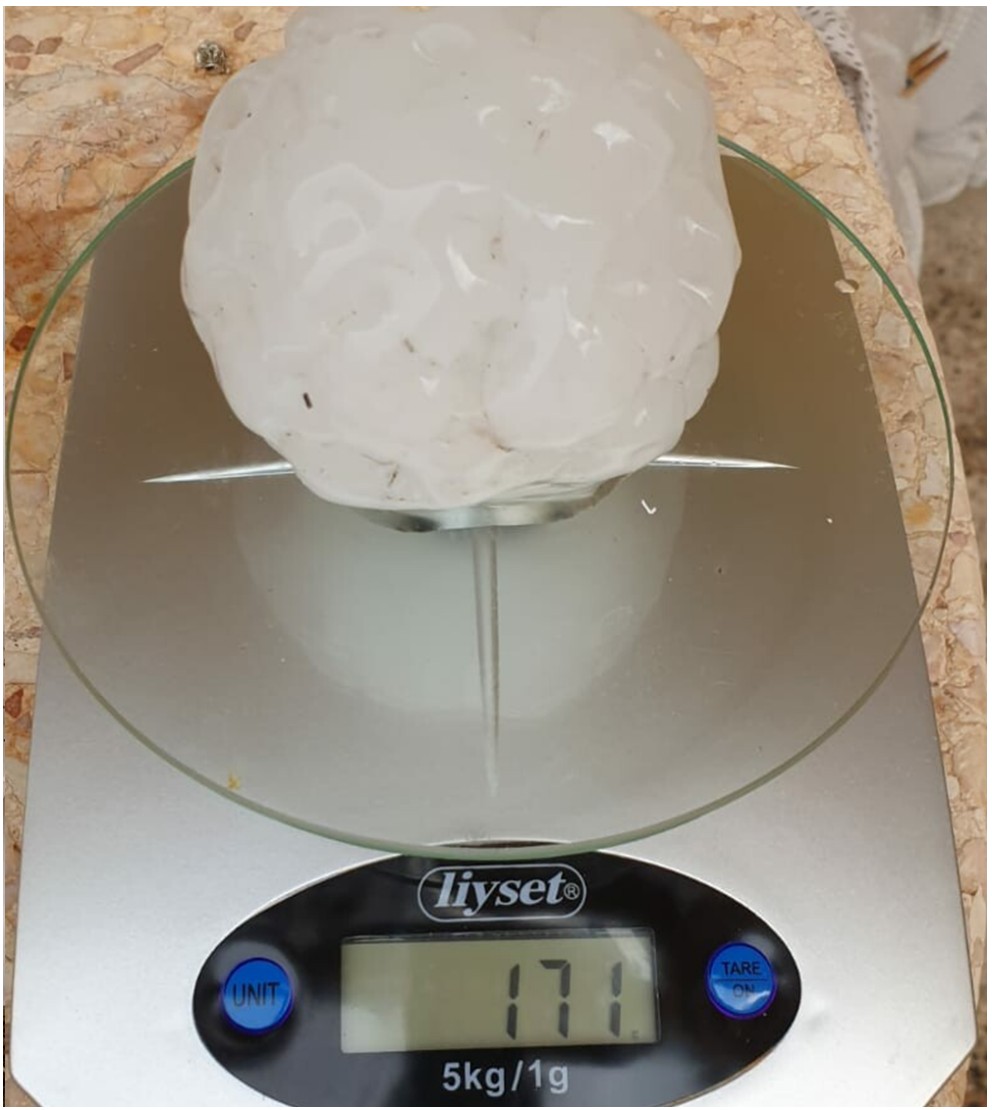

**Figure 4.** One of the largest hail-stones of the 30 August 2022 event (see the location in the red circle in the Figure 1). Credit: Angel Galan.

## 3. Results

### 3.1. Quality Control

The initial set of soundings included 17,504 profiles, considering that the radiosonde launching did not work on some days (6.8% of the 12-hourly periods did not have a sounding file). Table 2 presents the different categories obtained from the quality control procedure. The last class ("not valid sounding") considers twenty files that did not verify the conditions of the minimum size, sixty-eight files with not-readable records, and 926 files that have some non-solvable anomaly in one or more thermodynamic variables. In total, 5.4% files showed inconsistencies that made a not-valid sounding. Furthermore, 11.6% sounding files presented some minor repairable anomalies.

**Table 2.** Categorization of the sounding periods according to the quality control.

| Category | N Cases | % |
|---|---|---|
| No sounding | 1280 | 6.8 |
| Correct sounding | 14,312 | 76.2 |
| Sounding with minor anomalies | 2178 | 11.6 |
| Not valid sounding | 1014 | 5.4 |

Although there were some periods with a larger incidence in the sounding launching and triggering and the data acquisition, the distribution of discarded files is similar throughout the full period of data, except for three periods with a larger number of incidences: September 1997 to April 1999, May 2003 to July 2004, and November 2021 to January 2023. During these dates, the number of discarded soundings is slightly superior. In addition, there are no more than three days without valid soundings. Furthermore, all days with hail larger than 2 cm in diameter have had at least one sounding on the same day or the day before.

*3.2. Parameters Values in Hail Days*

Considering that SWDB goes from 2004 to 2022, both included, the total number of days was 6940. Of all these days, the number of hail days (considering any size) reported from the SWDB was 515 (7.4%). Limiting to a diameter equal or larger to 1 cm, the number of cases was 356 (5.1%). In the case of severe hail (diameter exceeding 2 cm), the number of days was 155 (2.2%), reducing to 40 days (0.6%) for very large hail (diameter over 4 cm).

Focusing on large and very large hail, Figure 5 shows how there has been a notable increase in days in the most recent years in both cases, with records in 2018 and 2022, respectively. However, although the length of the period is not long enough for considering climatically representative, the trend is evident. Another interesting point is the irregularity between years, with some cases without any occurrence surrounded by other years with a large number of days.

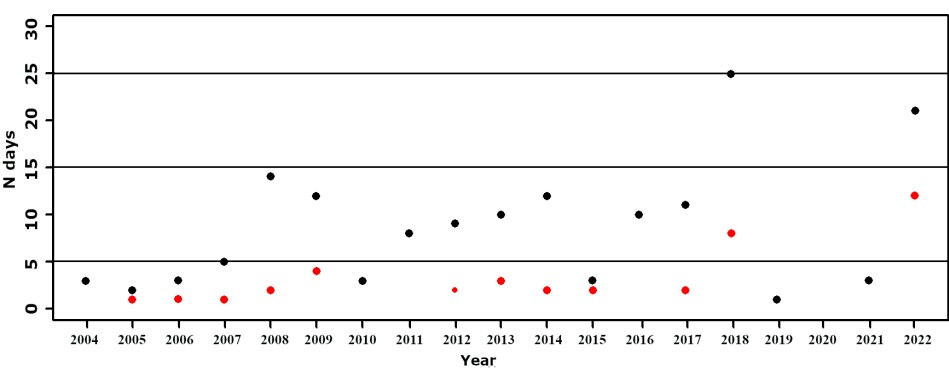

**Figure 5.** Number of days with large hail ($\geq$2 cm, black dots) and very large hail ($\geq$4 cm, red dots) for the period 2004 to 2022.

For each large and very large hail day, the mean and maximum values considering the period of the two days before and the day when hail occurred have been estimated. Figure 6 shows the violin plot for the six variables presented in Table 1. A violin plot is a combination of a box plot and a kernel density plot, allowing a better characterization of the distribution of the set value. In the present work, the white dot indicates the median value, the black box bottom and top show the first and third quantiles, and the grey area bottom and top locate the position of the minimum and maximum data-set values. In addition, the wider the plot at a given y-value, the more cases occurred at that level.

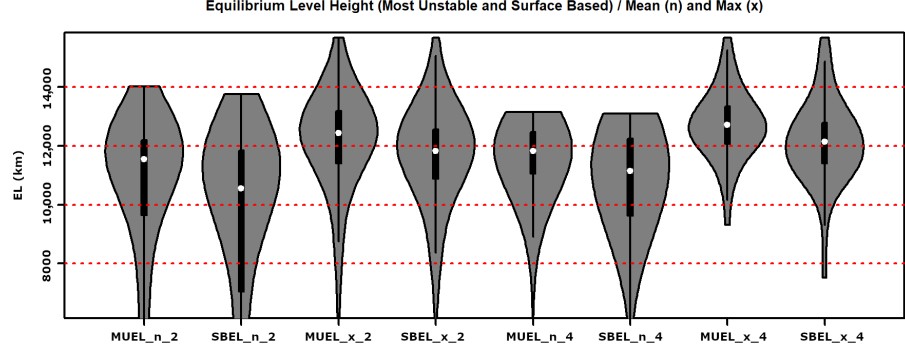

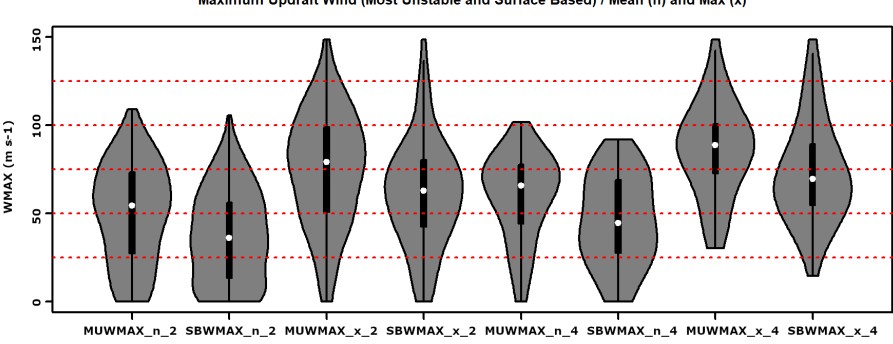

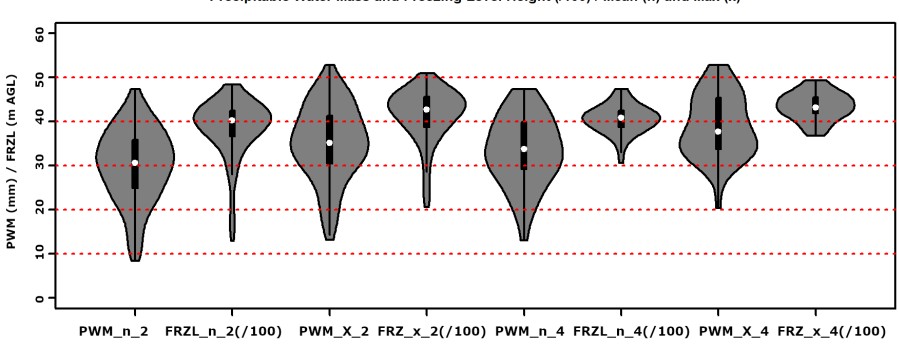

**Figure 6.** Violin plots of the selected parameters for the maximum (X) and mean (N) values for the two previous days and the current day of the hail event period. The parameters have been calculated for both large ($\geq$2 cm) and very large hail ($\geq$4 cm). From top to bottom: most-unstable (MU) and surface-based (SB) equilibrium level height (EL); most-unstable and surface-based maximum updraft speed (WMAX); and precipitable water mass (PWM) and freezing level height (FRZL, divided by 100).

Values for very large hail were higher than for large hail, but the differences were not critical. Focusing on the median (white dot), the equilibrium level height (most-unstable and surface-based) oscillated between 10,500 (surface-based for large hail) and 12,500 (most unstable for very large hail) m AGL.

Regarding the maximum updraft speed, the differences are similar to the EL: ranging between the maximum for the MU in the cases of very large hail (85 m s$^{-1}$) and the median for the SB in the large hail events (35 m s$^{-1}$). In any case, updrafts should be intense in those events with hail diameter exceeding 2 cm.

Finally, the PWM moved between 30 mm and 38 mm, indicating the necessity of moist environments in hail cases in Catalonia. On the other hand, the freezing level height has values similar to 4000 m AGL in practically all cases.

### 3.3. Trends of the Thermodynamic Parameters

Figure 6 reveals the positive trends of all the selected variables along the studied period (1998–2022, because 1997 and 2023 were incomplete). This trend is more positive in all cases for the 90th percentile, indicating how this increase impacted more in the extreme values. Considering that large and very large hail events usually occur with extremes, this could be indicative of the rise in the number of this type of hail cases.

Figure 7 and Table 3 summarize how the increase varies depending on the variable. The first feature is that the increase is larger in the 50th percentile in the case of the equilibrium level heights (both MU and SB), while in the rest of the variables, there is a higher increase in the 90th percentile. The variables with a major variation during the studied period were the MU vertical updraft, the precipitable water mass, and the MU equilibrium level height. On the contrary, the freezing level height suffered a negligible positive rise.

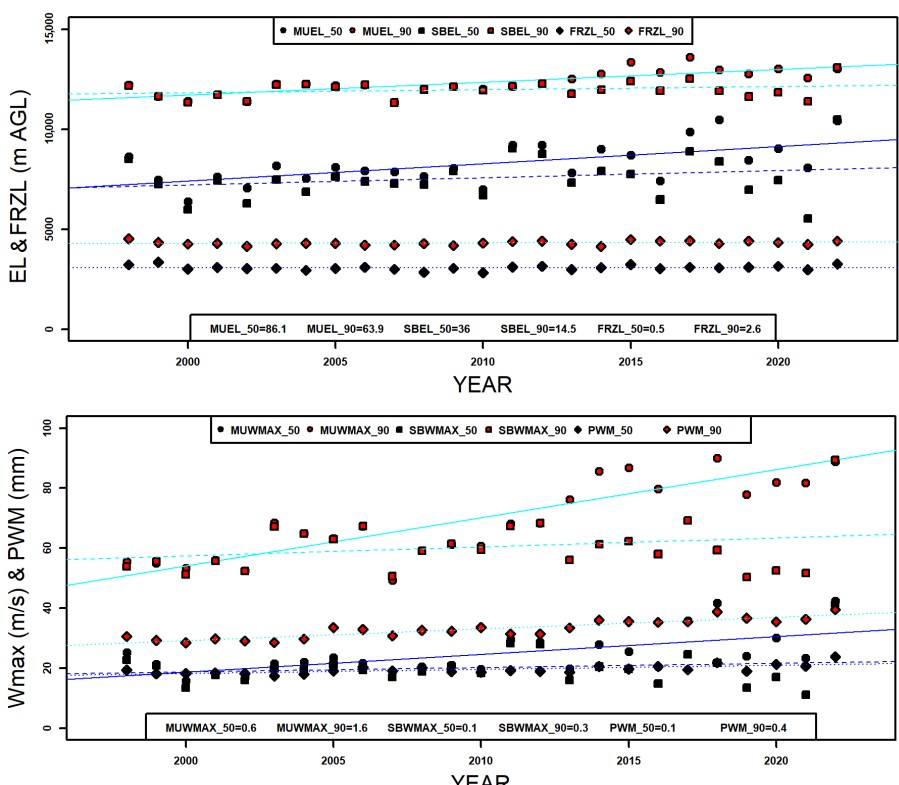

**Figure 7.** Yearly values and trends of the selected parameters for the 50th and 90th percentiles. Top: most-unstable (MU) and surface-based (SB) equilibrium level (EL) height and freezing level (FRZL) height; Bottom: most-unstable and surface-based maximum updraft speed (WMAX), and precipitable water mass (PWM). The lower legend box includes the rising rate according to the linear fitting model.

These results lead to two main points. First, parameters considering the virtual temperature (MUEL and MUWMAX) are more susceptible to being affected by environmental conditions changes. The higher increases of MUEL and MUWMAX in front of SBEL and SBWMAX coincide with other works found in the bibliography in this higher sensitivity [16,32,33]. In addition, the equilibrium level height changes have more influence in the severe hail-storm occurrence than freezing level height variations, probably because of the link with the higher development of the thunderstorms (also coinciding with the more intense updrafts). On the contrary, the changes in the melting layer dimensions have not directly influenced the hail events development.

**Table 3.** Rise percentage of the different variables with respect the maximum value.

| Variable | P50th | P90th |
|---|---|---|
| MUEL | 0.82 | 0.47 |
| SBEL | 0.34 | 0.11 |
| FRZL | 0.01 | 0.06 |
| MUWMAX | 1.41 | 1.58 |
| SBWMAX | 0.24 | 0.34 |
| PWM | 0.42 | 1.01 |

Finally, it is worth noting that the linear fitting presented in Figure 7 has high heterogeneity. There are some variables (MUEL, MUWMAX, and PWM) with adjusted R-squared moving between 0.6 and 0.8 and *p*-value notably under the threshold of 0.05. On the contrary, the other three variables (SBEL, SBWMAX and FRZL) presented adjusted R-squared values close to zero and high values of the *p*-value. This point indicates that the first group has better fitted to the data set. Furthermore, the results are statistically significant.

### 3.4. Was the August 2022 the Most Unstable Month of the Period?

To answer this question, it has compared the violin plot of the different variables for the full period (1998–2022, excluding the month of interest) with August 2022. Figure 8 shows this comparison, with the horizontal red line indicating the value for the month of interest.

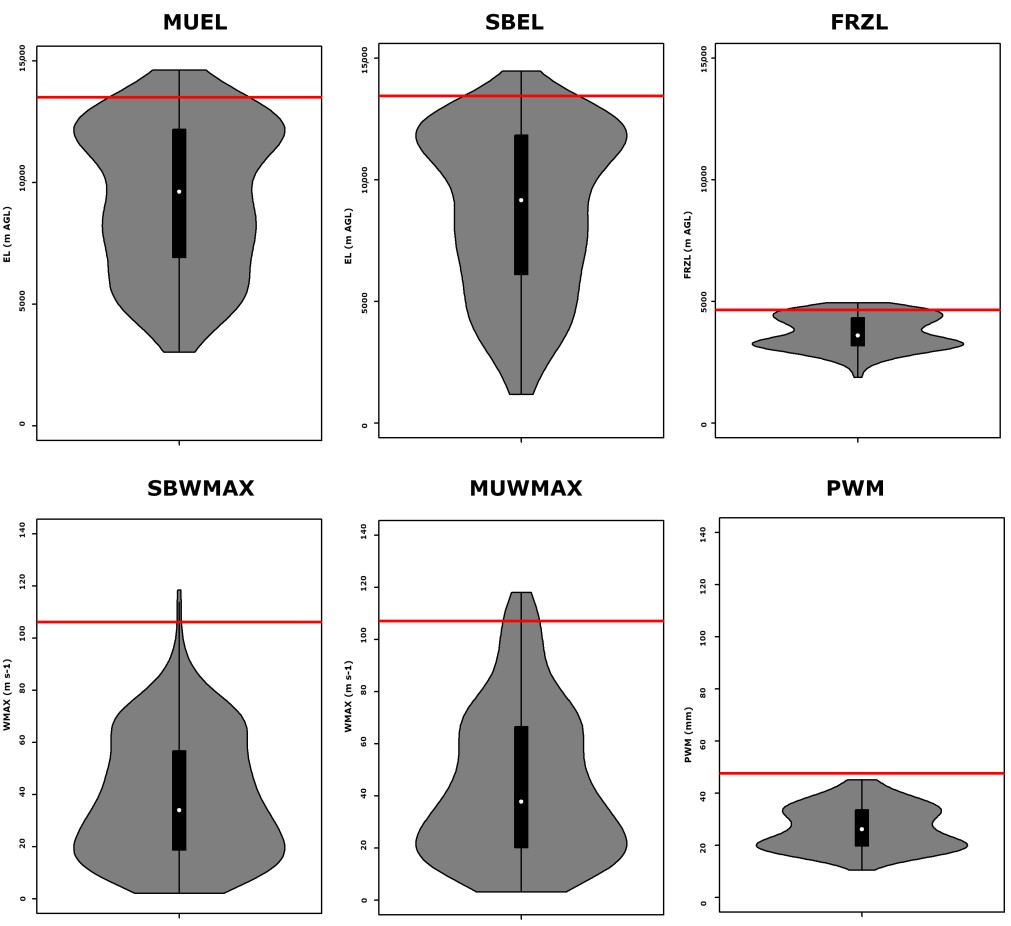

**Figure 8.** Violin plot for the whole period considering the monthly 90th percentile and excluding the August 2022 of the different parameters (from left to right and from top to bottom: MUEL, SBEL, FRZL, SBWMAX, MUWMAX, and PWM). The horizontal red line indicates the 90th percentile for the August 2022.

The values for August 2022 are extraordinary for the six parameters, always in the top part of the plots. In all cases, the line is clearly over 90% of the plot. In any case, two parameters stand out over the rest: the freezing level height, which was very close to the maximum value (4650 m vs. 4945 m), and the precipitable water mass, the unique variable that registered an absolute record (47.5 mm vs. 45.1 mm).

## 4. Discussion and Conclusions

The analysis of some thermodynamic variables along 1998–2022 and the comparison with the large and very large hail days in Catalonia has revealed some engaging results discussed with previous research. The analysis has focused on three points: Are the considered variables useful for hail diagnosis? Which are the trends of those variables during the studied period? What were the values of August 2022 following the hail that occurred (record of hail size and hail days) in Catalonia? In any case, the first point has been to determine if the Barcelona radio sounding is representative of the hail environments in Catalonia (an item previously treated by Gibergans [22] for heavy rainfall), considering that the region has 32,000 km$^2$ and a complex topography (see Figure 1). The results have shown that the answer is affirmative, and the combination of selected parameters provides a reasonable interpretation of the environment. However, it is necessary to go deeper into the discrimination of possible new candidates, according to the hail event nature (see, for instance [2]) and the region where the hail occurs. The discussion about the parameters' selection continues in the next paragraph.

The analysed variables selection was made according to the literature and considering the possibility of analysing the same variable (WMAX and EL) with two possible environments and taking advantage of the multiple options of library thundeR [30]: SB (surface-based) and MU (most-unstable). The differences between both options (MU and SB) indicate that the most-unstable variables seem more accurate for diagnosing hail occurrence. The evolution along the period is more coherent in the case of the MU, the same that occurred in the analysis of August 2022.

Two other variables have been considered: the precipitable water mass content (PWM) and the freezing level height (FRZL). The comparison with previous works in the region of study [2,24,28] indicates that the found values are significantly large: the WMAX was always over the previously proposed 50 m s$^{-1}$ (especially for the MU case), the PWM was 40 mm (instead of the 35 mm), and the freezing level height was close to 4000 m (while the previous threshold was 3000 m). However, it is important to indicate that these variations should link with the trend of the variables along the period, as previously cited. Furthermore, the thermodynamic variables have been estimated considering the virtual temperature correction, described in Taszarek et al. (2020) [36]. Then, new future research should be made to confirm the differences in the values and the influence of this correction. In particular, one would have to consider in the subsequent research, the effect on convection due to both orography and the particular thermodynamic situation at the time, the effect of which varies the type of instability and thus convection energetics.

Most of the research in the last years agrees in indicating that exists an increase in most of the values of the thermodynamic variables in the last years, which should continue in the future, according to different models [7–10]. These authors and many others also found that the local atmospheric conditions had a high weight in the positive or negative trends of hail occurrence in each region. In the case of Catalonia, the increasing conditions in the location of different heights (equilibrium and freezing level) and other variables (PWM and WMAX) have led to a rise in the number of severe hail events and in the size of the recorded hail stones (with the maximum in August 2022). Then, it seems that the local conditions can lead to determining that the same variables have a different effect on the hailstorm occurrence trends. Additionally, the future scenario forecast obtained in most of the previous research [7] in which the number of hail events will reduce does not agree with the observed in this analysis. The main reason could be that even exists a rising of the freezing level height (as in the other studies), the precipitable water mass and the vertical

updraft speed increase combination minimizes that the FRZL is higher than in the past. Then, the sea contribution can play a crucial role in the hail environment modelling in Catalonia, providing more convective energy and moisture to the atmospheric conditions.

The last point referred to the anomaly of August 2022, in which the number of large and very large hail days, and the hail size registered, broke all the previous records in Catalonia. The comparison of the selected parameter values for that month with the violin plots of the rest of the period (1998–2022) indicated that the month of interest presented extraordinary values. In all cases, the values were very similar to the maximum, but the most remarkable parameter was the PWM, which clearly exceeded the previous record. This parameter was previously cited by Farnell [2,24] as one of the indicators to diagnose and forecast large hail using thermodynamics in Catalonia.

Finally, some concluding notes are listed:

1   Thermodynamic variables provide helpful information for understanding atmospheric conditions. This fact also happens in a region with a high topographic influence, as in the case of Catalonia. The equilibrium and freezing level heights, the maximum updraft speed, and the precipitable water mass can forecast an environment prone to hail storms. However, these conditions are necessary but not always enough for hail occurrence. The other hail-occurring modelling element corresponds with micro-physical factors, which are more unpredictable and can inhibit hail production.

2   The selection of the variables is highly dependent on the region. It is crucial to know the thermodynamic conditions of the zone. It is advisable to carry out a preliminary study or, on the other hand, to consider the previous bibliography if it exists. Furthermore, some of them (with high similitude) can provide very different results. This circumstance is the case of the MU EL or the SB EL.

3   The selected variables have adequately described the changes in the hail occurrence in the region in this research. However, these results do not agree with different analyses in other zones. This point concludes that the sea contribution is relevant in the Catalan case, but the Mediterranean Sea effects could be different in other basin points.

4   The PWM has provided the best correlation with the hail occurrence, followed by the MUEL and the MUWMAX. On the contrary, the FRZL provided the worst results compared with the hail events. Then, the relation between FRZL rise and the decrease in the number of cases observed in other areas can be counteracted mainly by the increase in the PWM.

5   The positive trend observed in all analysed variables has shown different slopes, with more relevant increases in MUEL, MUWMAX and PWM. On the contrary, the FRZL and SB variables increased slightly over the period.

6   The PWM was the variable with the more relevant values during the August 2022 episode, compared with the global registers of the studied period. This variable broke the previous record, with values never estimated in the analysed period. This fact could partially explain the exceptional hail events of that month.

These conclusions can be summarized in two specific considerations:

-   From the physical point of view, the variables with values exceeding the 25-years period were the freezing level height and the precipitable water mass. On the contrary, the equilibrium level and the maximum updraft speed had values over the third quartile but were far from the historical record. FRZL indicates the altitude at which the particles start to change to a solid state. Although their values were exceptional and should be an inhibitor factor of the hail generation [7,15], they were balanced by the high quantities of moisture in the atmosphere and the intense vertical updrafts.

-   From the climatic point of view, the positive trend of all the analysed variables agrees with most of the previous studies [7,8,10,16,24,37]. However, the contribution of the PWM looks more relevant than in other regions. This point could explain why the hail events number has risen in Catalonia in the last few years. Finally, it is crucial to note



that thermodynamics cannot explain all factors and that it is necessary to combine it with other elements, such as synoptic studies [18].

**Funding:** This research received no external funding.

**Institutional Review Board Statement:** Not applicable.

**Informed Consent Statement:** Not applicable.

**Data Availability Statement:** The data presented in this study are available on request from the corresponding author.

**Acknowledgments:** The author wants to thank the Servei Meteorològic de Catalunya and the Physics Faculty of the Universitat de Barcelona for the data provided, and to Angel Galan for the information regarding the 30 August 2022 event.

**Conflicts of Interest:** The author declare no conflict of interest.

## Abbreviations

The following abbreviations are used in this manuscript:

| | |
|---|---|
| CAPE | Convective Available Potential Energy |
| ASL | Above Sea Level |
| NE | North-East |
| LI | Lifted Index |
| PWC | Precipitable Water Content |
| QC | Quality Control |
| UTC | Universal Time Coordinate |
| MU | Most-unstable |
| EL | Equilibrium Level |
| SB | Surface-based |
| WMAX | Maxim updraft speed |
| PWM | Precipitable Water Mass |
| FRZL | Freezing Level Height |
| AGL | Above Ground Level |
| SMC | Servei Meteorològic de Catalunya |
| SWDB | Severe Weather Database |
| RSD | Radio-sounding base location |

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
