# Peer review of "Thermodynamic Conditions during August 2022 in Catalonia: The Monthly Record of Hail Days, Hail Size and the Differences in the Climatic Values"

_climate, doi:10.3390/cli11090185_

Round 1
Reviewer 1 Report
The paper is discussing research conducted on a hailstorm that occurred on August 30, 2022, in the northeastern part of Catalonia, notable for producing unusually large hailstones, with diameters exceeding 10 cm. The study also found that hail events with diameters exceeding 2 cm occurred on fourteen days and those exceeding 4 cm occurred on seven days during August 2022. These observations set new records for hail size and frequency in Catalonia over the past three decades, The paper compares the statistics of thermodynamic values obtained from atmospheric soundings in Barcelona with climatic data spanning from 1998 to 2022 (a 25-year period)in order to assess the behavior of these thermodynamic parameters during hail events in Catalonia. Among the parameters considered, precipitable water mass emerged as the most effective predictor of hail occurrence.
This part of the study is the most interesting, because it directly correlates moisture fluxes and the thermodynamic situation. Given the abundance of data available in Catalonia a study that correlates thermodynamic variables with the mechanisms of convection that lead to such explosive phenomena as hail development outside the statistical parameters of the region seems to be very interesting. The paper poses these questions at the end of the discussion lines 200-209.
In particular, one would have to consider in the subsequant research, the effect on convection due to both orography and the particular thermodynamic situation at the time, the effect of which, varies the type of instability and thus convection energetics.
L103-104 In addition, there are no more than three days without valid soundings.
L126-127 In addition, the wider the plot at a given y-value, the more cases occurred at that level.
L251-253 It is advisable to carry out a preliminary study or, on the other hand, to consider the previous bibliography if it exists.
Furthermore, some of them (with high similarity) can give very different results.
L265-266 On the contrary, the FRZL and SB variables increased slightly over the period.
L282-284 Finally, it is crucial to note that thermodynamics cannot explain all factors and that it is necessary to combine it with other elements, such as synoptic studies.
Author Response
Dear Reviewer,
Thank you very much for your proposals. Please, find attached my answers and the new version, considering your interesting suggestions.
Best regards

Reviewer 2 Report
Review for Article entitled:
Thermodynamic conditions during August 2022, the monthly record of hail days and hail size, in Catalonia) NE of the Iberian Peninsula) and the differences in the climatic values.
By Rigo Tomeu
Comments
l The article is dealing with hail occurrence and hail size, examining the hailstorm thermodynamic environment produced large hail with big hail size in record. A set of basic thermodynamic parameters are examined to compare the differences in hail size. The results are a useful contribution in studying specific atmospheric parameters related to hail size and in this frame the paper can be accepted for publishing after considering some suggestions in the following.
l Some correction is needed in the title, since applying of parentheses and dots are not used in titles.
l Some more details should be given for the giant hail event in record or perhaps a photo should be provided, i.e. exact area recorded, time of the event, etc.
l The reference no [1] should be written and described in a more detail, i.e. where officially is published and can be addressed.
l In “Introduction” some information should be included about hail occurrence and hail size if they are reported from individuals and how these reported are confirmed and verified.
l In line 54, the period is used to be written 1998-2022 instead of 1998ˇ2022 and should be changed everywhere in the text.
l In line 59, some information should be added in the Barcelona sounding data used are from twice a day, for 00:00 and 12:00 UTC
l In the reference no [28], some more details should be provided, i.e. page numbers in the Proceedings.
l Figure 4 should be corrected to be more clearly readable, i.e. all the years should be displayed in the x-axis. The dot size should be also rather increased.
l In page 12, the concluding note no 1, seems gives a controversial meaning about the usefulness of the parameters mentioned for forecasting or not hail occurrence and should be eliminated or rewritten.
l In page 12, the concluding note no 2 seems not necessary to be included since the purpose of the study is not to evaluate “ThundeR software”.
l Overall, the paper needs sme English language improving in both syntax and appropriate phraseology.
Author Response

(The authors gave the same response as above.)

Reviewer 3 Report
The manuscript analysed the case study of extreme hailstorm over the Iberian Peninsula and compared the thermodynamic indices with the climatological values. Overall, the manuscript is well written and making its point with adding information about the extreme event to the scientific community. I have few suggestions for the further improvements of the work before publication.
1. The introduction section needs more discussion on the use of thermodynamic indices from the radiosonde observations. There are various studies dealing with hailstorms, thunderstorms and lightning around the globe.
2. Kindly explain why the use of thermodynamic indices is important and how this study will be helpful in prediction or identification of such events in future. Validating previous studies or trend determination is fine, but can not be presented as the objective of the work.
3. Please provide the latitude and longitude on the x and y axis of Figure 1.
4. Can you present the result of quality check files in table format. Although you mentioned the number of incomplete files after the quality check, it is not very clear, how many of these incomplete files had hailstorm reported.
5. Please explain with formulae, what are the difference between the most unstable and surface-based parameters for the plots.
6. Figure 6 top panel, the y-axis legend needs to be changed as it is currently displaying a ratio between two variables.
7. Kindly elaborate the findings of Table 2.
8. Please explain the logic behind selecting only MUEL, MUWMAX, PWM, SBEL, SBWMAX and FRZL for the current study beside many other available indices.
Author Response

(The authors gave the same response as above.)
